

# Specimen alignment with limited point-based homology: 3D morphometrics of disparate bivalve shells (Mollusca: Bivalvia)

Stewart M. Edie[1], Katie S. Collins[2] and David Jablonski[3,4]

[1] Department of Paleobiology, National Museum of Natural History, Smithsonian Institution, Washington, DC, United States
[2] Department of Earth Sciences, Invertebrates and Plants Palaeobiology Division, Natural History Museum, London, United Kingdom
[3] Department of the Geophysical Sciences, University of Chicago, Chicago, IL, United States
[4] Committee on Evolutionary Biology, University of Chicago, Chicago, IL, United States

## ABSTRACT

**Background:** Comparative morphology fundamentally relies on the orientation and alignment of specimens. In the era of geometric morphometrics, point-based homologies are commonly deployed to register specimens and their landmarks in a shared coordinate system. However, the number of point-based homologies commonly diminishes with increasing phylogenetic breadth. These situations invite alternative, often conflicting, approaches to alignment. The bivalve shell (Mollusca: Bivalvia) exemplifies a homologous structure with few universally homologous points—only one can be identified across the Class, the shell 'beak'. Here, we develop an axis-based framework, grounded in the homology of shell features, to orient shells for landmark-based, comparative morphology.

**Methods:** Using 3D scans of species that span the disparity of shell morphology across the Class, multiple modes of scaling, translation, and rotation were applied to test for differences in shell shape. Point-based homologies were used to define body axes, which were then standardized to facilitate specimen alignment *via* rotation. Resulting alignments were compared using pairwise distances between specimen shapes as defined by surface semilandmarks.

**Results:** Analysis of 45 possible alignment schemes finds general conformity among the shape differences of 'typical' equilateral shells, but the shape differences among atypical shells can change considerably, particularly those with distinctive modes of growth. Each alignment corresponds to a hypothesis about the ecological, developmental, or evolutionary basis of morphological differences, but we suggest orientation *via* the hinge line for many analyses of shell shape across the Class, a formalization of the most common approach to morphometrics of shell form. This axis-based approach to aligning specimens facilitates the comparison of approximately continuous differences in shape among phylogenetically broad and morphologically disparate samples, not only within bivalves but across many other clades.

Corresponding author
Stewart M. Edie, edies@si.edu

# INTRODUCTION

Comparative morphology depends on how organisms are oriented, or aligned. For a simplistic example, a kiwi's beak is relatively long for a bird when measured from the tip to the base of the skull, but rather short when measured from the tip to the nostrils (an alternative definition of beak length; *Borras, Pascual & Senar, 2000*). Thus, the choice of anatomical reference points can profoundly alter our interpretations of evolutionary morphology. Alignments commonly use point-based aspects of homologous features—the junction of the kiwi's beak with the cranium (a Type I landmark; *Bookstein, 1991*) and the distal-most point of the beak, the tip (a Type II landmark). Closely related organisms tend to share more of these homologous points, allowing for a straightforward alignment and comparison of their shapes. Alignment on strict, point-based homology becomes more problematic with increasing phylogenetic distance, as the number of homologous features invariably diminishes (*Bardua et al., 2019*).

Bivalve mollusks have become a model system for macroevolution and macroecology (*Jablonski et al., 2017*; *Edie, Jablonski & Valentine, 2018*; *Crame, 2020*), but their strikingly disparate body plans complicate Class-wide morphological comparisons using strict homology (*Cox, Nuttall & Trueman, 1969*; cf. *Chione* and *Pecten* in Fig. 1). Inimical to triangulation and thus alignment *via* landmarks, the valve of the bivalve shell—the most widely accessible feature of the animal for extant species in museum collections and throughout the fossil record—has only one homologous point: the apex of the beak, which is the origin of growth of the embryonic shell (*Carter et al., 2012*:21; Figs. 1A, 1D). Homology-free approaches can be useful for comparing the shapes of shell valves when anatomical orientation is either unknown or uncertain (*Bailey, 2009*); but wholesale substitution of shape, *i.e.* analogy, for homology complicates the evolutionary interpretation of morphological differences. Despite the lack of multiple homologous *points* on the shell valve across the Class, a number of its *features* are homologous and can facilitate comparisons. Following an overview of previous approaches to orienting shells, we apply principles of bivalve comparative morphology to develop a framework for aligning shell valves (hereafter 'shells') across the Class, thus enabling phylogenetically extensive analyses of their shapes despite their remarkable range of body plans.

## Approaches to orienting the bivalve shell

Many body directions, axes, lines, and planes have been defined for bivalves (see *Cox, Nuttall & Trueman, 1969*; *Bailey, 2009*; *Carter et al., 2012*)—some related to features of the shell (an accretionary exoskeleton composed of calcium carbonate), and others to features of the soft body (the digestive tract, foot, byssus, muscles, etc.). Separation into these 'shell' and 'body' terms is a false (*Stasek, 1963a*) but convenient dichotomy (*Yonge, 1954*): the shell is generated by, and remains attached to, the soft body, but their morphologies can become decoupled (*Yonge, 1954*; *Edie et al., 2022*). Still, both shell and body features are required for orientation *via* homology (*Stasek, 1963a*; cf. *Bailey, 2009*). Given our goal of

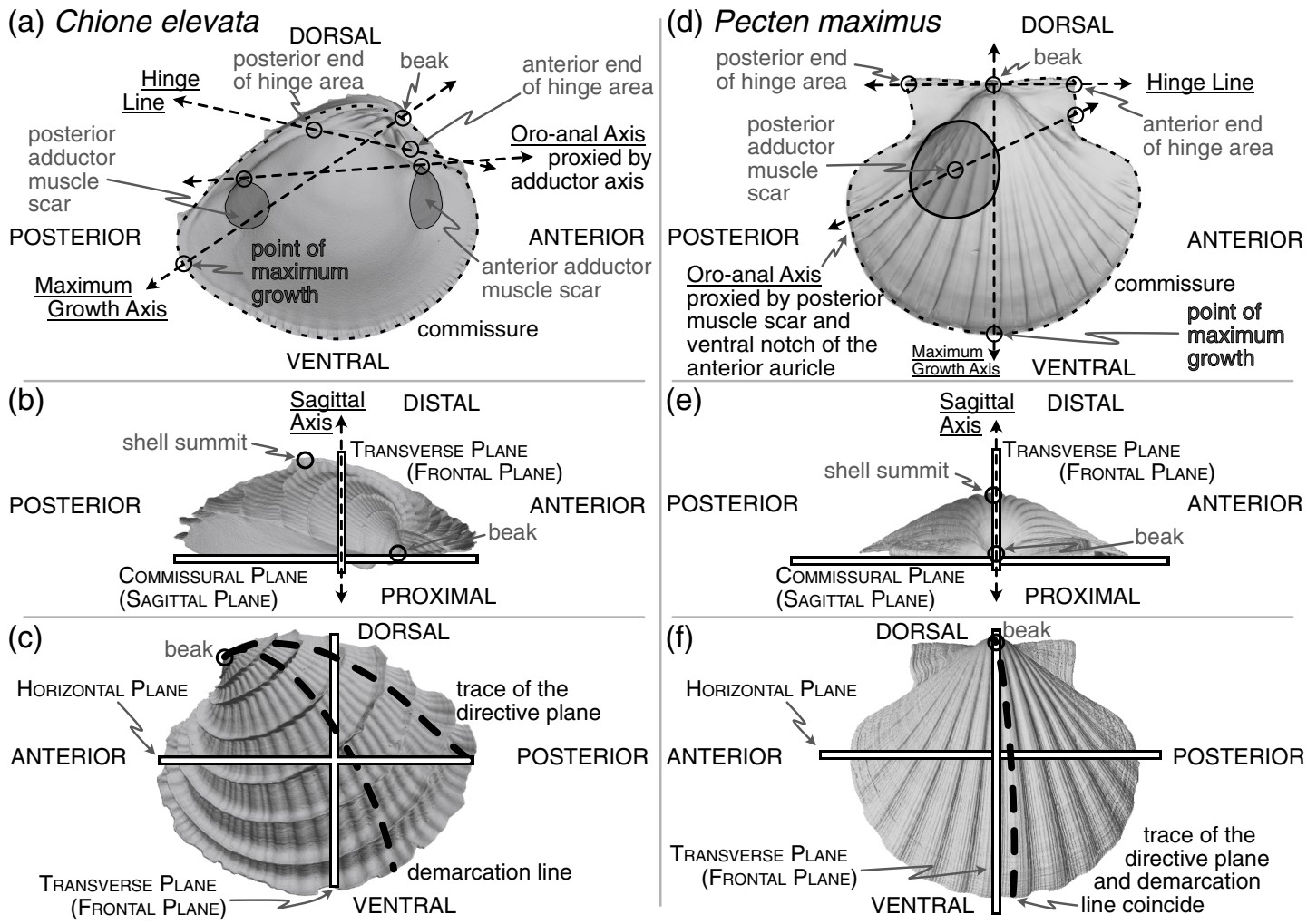

**Figure 1 Positions of shell features, axes, and planes as mentioned and defined in the main text.** (A) An interior view across the commissural (sagittal) plane for a helicospiral shell *Chione elevata* (Say 1822), with marked positions of the beak, commissure, point of maximum growth along the commissure, posterior and anterior ends of the hinge area, oro-anal axis, maximum growth axis, and hinge line. (B) An exterior view of *C. elevata* across the horizontal plane, with marked positions of the beak, shell summit, sagittal axis, transverse (frontal) plane, and commissural (sagittal) plane. (C) An exterior view of *C. elevata* across the commissural plane, with marked positions of the beak, trace of the directive plane, demarcation line, horizontal plane and transverse plane. (D–F) As in panels A-C, but for a more planispiral shell *Pecten maximus* (Linnaeus 1767).

aligning the shell for quantitative, comparative morphological analysis across broad phylogenetic scales and through the fossil record, we focus on orientations that can be inferred from this part alone, but we must use a critical body axis to fully determine orientation, the anteroposterior axis. Thus, shell features used for alignment can be divided into the three classes considered below: (1) intrinsic characteristics of the shell relating to its geometry and growth, (2) the shell's biomechanics, and (3) proxies of the body recording the positions of the soft internal anatomy (hereafter 'soft body') including the adductor muscle scars, pallial line, byssal notch, pedal gape, siphon canal, and more.

## Orientation *via* shell geometry

The only homologous point on the shell across all bivalves is the apex of the beak (hereafter 'beak', Figs. 1A, 1D), which has led to approaches that use other aspects of the shell's geometry to orient specimens. However, these lines and planes of shell geometry have been criticized for their lack of homology (*Stasek, 1963b*:226) and ubiquity across the class (*Lison, 1949*:62; *Owen, 1952*:149; *Carter, 1967*:272). We consider two such features in this section, the directive plane and the demarcation line (Figs. 1C, 1F and defined below), before proposing two alternative means of orienting shells that draw on the shell's geometry and growth: the maximum growth axis and shape of the shell commissure (Figs. 1A, 1D).

Related to the geometry and growth of the shell, the **directive plane** (*Lison, 1949*) was proposed as the only plane passing through the shell that contains the logarithmic planispiral line connecting the beak to a point on the ventral margin (Figs. 1C, 1F); all other radial lines would be logarithmic turbinate spirals (or 'helicospirals'; *Stasek, 1963b*:217). In other words, on a radially ribbed shell, there may be a single rib that lies entirely on a plane when viewed from its origin at the umbo to its terminus on the ventral margin; that plane is orthogonal to the commissural plane for planispiral shells (*e.g.* many Pectinidae, Fig. 1F), but lies at increasingly acute angles to the commissural plane with increasing tangential components of growth (*i.e.* geometric torsion; see the trace of the directive plane on *Chione* in Fig. 1C and examples in *Cox, Nuttall & Trueman, 1969*:86-figs. 70–71). In theory, the directive plane could be used to orient the dorsoventral axis of the shell, but in practice, the feature is not universal across shell morphologies (*e.g.* the strongly coiled *Glossus humanus* (as *Isocardia cor*) in *Lison, 1949*:62; *Owen, 1952*:149). *Cox, Nuttall & Trueman (1969*:87) also remark that the plane cannot "be demonstrated easily by visual inspection if the shell lacks radial ribbing, except in rare specimens with an umbonal ridge that proves to lie within the directive plane." Difficulty in application is a poor basis for avoiding an approach, but the non-universality of this feature renders it inapplicable to Class-wide comparisons of shell shape.

*Owen (1952)* proposed an alternative to the directive plane: the **demarcation line** (Figs. 1C, 1F; originally termed the 'normal axis' but re-named by *Yonge, 1955*:404). As with the directive plane, the demarcation line serves to orient the dorsoventral direction and separate the shell into anterior and posterior 'territories' (*Yonge, 1955*:404; *Morton & Yonge, 1964*:40), but its definition has been variably characterized in geometric and/or anatomical terms. Per *Owen (1952*:148), the demarcation line can "be considered with reference to three points: the umbo, the normal zone of the mantle edge and the point at which the greatest transverse diameter of the shell intersects the surface of the valves." *Yonge (1955*:404), acknowledging correspondence with Owen, described the demarcation line as: "the projection onto the sagittal plane of the line of maximum inflation of each valve … starting at the umbones. … *i.e.* the region where the ratio of the transverse to radial component in the growth of the mantle/shell is greatest." *Carter et al. (2012*:52) provided perhaps the clearest description as the line defining the "dorsoventral profile when the shell is viewed from the anterior or posterior end." However, *Stasek (1963b)*

demonstrated the difficulty in measuring this line; note the nearly orthogonal orientations of the empirically determined demarcation line on *Ensis* (*Stasek, 1963b*:225-fig. 5A) compared to its initially proposed position (*Owen, 1952*:148-fig. 5). Stasek's empirical approach, coupled with the revised definition of Carter et al., is tractable with today's 3D-morphology toolkit. But, critically, this definition depends on the direction of the anteroposterior axis, which itself is variably defined (see discussion in next section). Thus, definitionally driven shifts in the direction of the anteroposterior axis can alter the trace of the demarcation line. Owen's initial definition is independent of the anteroposterior axis, but as Stasek demonstrated, its identification can be unreliable. Thus, high degrees of digitization error for the demarcation line may confound comparisons of shell shape, and we do not recommend the demarcation line as a feature for aligning shells across the Class.

Both the directive plane and the demarcation line attempt to orient the shell on aspects of its geometry that are tied to its growth. A similar and more reliably determined approach may be orientation to the **maximum growth axis** (*i.e.* 'line of greatest marginal increment' *sensu Owen, 1952*; Figs. 1A, 1D). The maximum growth axis is the straight line that connects the origin and terminus of the trace of maximum growth across the shell surface. This trace connects the beak to the ventral margin along a perpendicular path to the most widely spaced commarginal growth increments (as such, this definition appears to have similar properties to the trace of the directive plane on the shell surface). But, as for the directive plane and the demarcation line, the maximum growth axis can be prone to measurement error without fitting a formal model of shell growth (*e.g.* those of *Savazzi, 1987*; *Ubukata, 2003*), and should therefore be used with caution. However, a reasonable and reliably measured proxy for this axis is the line lying on the commissural plane that originates at the beak and terminates at the furthest point on the shell commissure. Thus, this axis can indicate the dorsoventral orientation of the shell.

Orientation using the **shape of the shell commissure** offers, arguably, the most reliably determined approach that uses shell geometry (Figs. 1A, 1D). Given the accretionary growth of the shell, points on the commissure—the homologous leading edge of shell growth (*Vermeij, 2013*)—are geometrically homologous, or correspondent (*Bookstein, 1991*; *Gunz, Mitteroecker & Bookstein, 2005*). Valve handedness is still required to ensure that compared valves are from the same side of the body (*i.e.* left *vs.* right), which requires anteroposterior directionality (see below). This alignment thus orients shells using geometric correspondence based on homology of growth; it draws on the same homology of the growing edge as *Raup (1966)* shell coiling models, which were extensions of the concepts developed through the directive plane and demarcation line.

The **sagittal axis** is crucial to the shell's three-dimensional alignment, and is likely the least controversially defined. This axis is the pole (= normal) to the sagittal plane, which lies parallel to the commissural plane defined as: "the more proximal part of the line or area of contact of the two shell valves" (*Carter et al., 2012*:38). Therefore, the sagittal axis is parallel to the frontal and horizontal planes (Figs. 1B, 1E). The proximal direction is towards the commissural plane and the distal direction is towards the shell's summit: the point on the shell that is maximally distant from the commissural plane (Figs. 1B, 1E, *Cox, Nuttall & Trueman, 1969*:108; *Carter et al., 2012*:177). If valve handedness (*i.e.* left *vs.*

right laterality) and the directionality of the dorsoventral and anteroposterior axes are known, then this axis is rarely required for orientation. However, certain definitions of the anteroposterior and dorsoventral axes are not constrained to be orthogonal (*e.g.* in monomyarian taxa, such as Pecten in Fig. 1D, with the anteroposterior axis defined as the oro-anal axis, see below, and the dorsoventral axis as the axis of maximum growth); as the axes representing the anteroposterior and dorsoventral directions become more parallel, then the sagittal axis becomes an increasingly important safeguard against the inversion of the proximal-distal direction in quantitative alignments.

## Orientation *via* shell biomechanics

The hinge has been treated as a "fixed dorsal region" (*Yonge, 1954*:448; see also *Jackson, 1890*:282), later redefined to reflect the position of the mantle isthmus bridging between the two valves as a universally dorsal-directed feature (*Cox, Nuttall & Trueman, 1969*:79). Beyond its determination of dorsoventral directionality, the hinge, specifically the hinge axis defined as the "ideal line drawn through the hinge area, and coinciding with the axis of motion of the valves" (*Jackson, 1890*:309), is a Class-wide feature that can constrain one of the three Cartesian axes required for alignment. In a strictly mechanical sense, the ligament, and not the hinge teeth, directs the orientation of the axis of motion (*Trueman, 1964*:56; *Cox, Nuttall & Trueman, 1969*:47; *Stanley, 1970*:47). However, the hinge area, which includes both the ligament and teeth (see 'hinge' in *Carter et al., 2012*:74 and Figs. 1A, 1D), is hypothesized to be analogous in function—guiding the two valves into alignment during closure (*Cox, Nuttall & Trueman, 1969*:47)—and homologous in its origin (*Scarlato & Starobogatov, 1978*; *Waller, 1998*; *Fang & Sanchez, 2012*).

For quantitatively aligning shells, the **hinge line** is determined by the two farthest apart articulating elements of the hinge area, excluding lateral teeth, which are variably present among heterodont species (*e.g. Mikkelsen et al., 2006*:493; *Taylor & Glover, 2021*); the definition proposed here is a synthesis of the discussions in *Cox, Nuttall & Trueman (1969*:81) and *Bradshaw & Bradshaw (1971)*. Thus, by directing the orientation of the horizontal plane, which divides the body into dorsal (towards the beak) and ventral (towards the free edge of the shell) territories, the hinge line can also proxy the anteroposterior axis (Figs. 1A, 1D; *Cox, Nuttall & Trueman, 1969*:81 and discussion below).

## Orientation *via* indicators of the soft body on the shell

Anteroposterior directionality is the third Cartesian axis required for orienting the bivalve shell in three-dimensions. The positions of the mouth (anterior) and anus (posterior) ultimately determine the anteroposterior axis (*Jackson, 1890*; 'preferably' described as the 'oro-anal' axis in *Cox, Nuttall & Trueman, 1969*:79), but the exact positions of these two soft-body features are rarely recorded directly on the shell. Thus, tactics for determining the polarity, if not the precise bearing, of the anteroposterior axis have relied on proxies specific to lineages or body-plans—shell features that are assumed to correlate with positions of the soft-body (*e.g.* positions of the adductor muscle scars, Figs. 1A, 1D; *Cox, Nuttall & Trueman, 1969*:79). Disparate body plans necessitate taxon-specific rules for

orientation, such that determining the anterior and posterior ends of the shell requires a mosaic approach. For example, there are at least three definitions of the anteroposterior axis in dimyarians alone (*Bailey, 2009*:493), which necessarily differ from those of monomyarians given the presence of two, instead of one, adductor muscle scars. For those monomyarians, which commonly have lost the anterior adductor (*Yonge, 1954*; but see loss of the posterior adductor in the protobranch Nucinellidae *Allen & Sanders, 1969*, *Glover & Taylor, 2013*), additional shell features are used to orient the anteroposterior axis. In pectinids, the byssal notch of the anterior auricle proxies the location of the mouth (Fig. 1D), but in ostreids, the mouth is more centrally located under the umbo, near the beak (*Yonge, 1954*:448).

Lineage or body-plan specific definitions help with anteroposterior orientation of shells that lack point-based homology (*e.g.* two muscle scars *vs.* one), but they still rely on proxies for the position of soft-body features that may not be determined for taxa known only from their shells, *e.g.* some fossils (*Bailey, 2009*). Hypothesizing anteroposterior orientation using phylogenetic proximity to extant clades may help, but this approach should be used with caution in given the lack of direct anatomical evidence—especially when phylogenetic affinities are either unknown or distant, as for many Paleozoic taxa (*Bailey, 2009*). Nor is it advisable to assume that the anteroposterior axis is identical to another, well-defined axis such as the hinge line (see variable bearings of the anteroposterior and hinge axes in *Cox, Nuttall & Trueman, 1969*:80-fig. 64). However, if the phylogenetic or temporal scope of an analysis precludes the determination of the anteroposterior axis using homologous body features with geometric correspondences (*e.g.* inclusion of dimyarian and monomyarian taxa), then the hinge line can be used as a Class-wide proxy; then, multiple, taxon-specific features can be used to determine the anterior and posterior ends of the shell (*Cox, Nuttall & Trueman, 1969*).

## Alignment of shells for comparative morphological analysis: proposed protocol

The challenge is to reconcile the different means of orienting bivalve shells discussed above to specify three universal axes across the Class for comparative morphometrics in three dimensions. Here, we compare the differences in shell shape produced by five alignment schemes listed below. In all alignments, the sagittal axis determines the lateral orientation of the shell (*i.e.* the less commonly studied 'width' dimension). The anteroposterior and dorsoventral directions vary according to the direction of a chosen anatomical axis or line. Precise definitions of landmark placement for each axis are provided in the Methods.

- **SX-HL-oHL.** Anteroposterior orientation determined by the hinge line (HL); dorsoventral orientation determined by the orthogonal line to the HL within the commissural plane (oHL). This alignment emulates the orientation scheme for measuring shell height, length, and width—the most common and widely applicable framework for comparing shell morphology (*Cox, Nuttall & Trueman, 1969*:81–82; *Kosnik et al., 2006*).

- **SX-OAX-oOAX.** Anteroposterior orientation determined by the proxied positions of the mouth and anus using shell features (oro-anal axis, OAX); dorsoventral orientation determined by the orthogonal line to the OAX within the commissural plane (oOAX). Similar to SX-HL-oHL, this alignment largely determines orientation by a single axis, the OAX, which has also been used to frame linear measurements of shell morphology (*e.g. Stanley, 1970*:19).
- **SX-HL-GX.** Anteroposterior orientation determined by the hinge line (HL); dorsoventral orientation determined by the maximum growth axis (GX). This alignment allows an aspect of shell growth to affect its orientation and thus the degrees of morphological similarity among specimens.
- **SX-HL-GX-OAX.** Anteroposterior orientation determined by the directions of both the HL and OAX; dorsoventral orientation determined by GX. This 'full' alignment scheme incorporates axes derived from intrinsic characteristics of the shell and the soft body for orientation. For example, the HL and OAX are not necessarily congruent, and thus both axes can contribute to anteroposterior orientation.
- **SX-COMM.** Anteroposterior and dorsoventral orientation determined by the shape of the commissure curve, with the initial point nearest the beak (Fig. 2A). This alignment uses the geometric correspondence of semilandmarks on the commissure that capture the relationship between its shape and growth.

To compare the effects of alignment choice on the differences in shell shape, we adopt the procedure for Procrustes superimposition—translation to a common origin, scaling to a common size, and rotation to minimize relative distances of landmarks (*Zelditch, Swiderski & Sheets, 2012*). While we are most concerned with assessing the effects of rotation using the five alignments described immediately above, choices of translation and scaling can also influence shape differences. Thus, we consider all combinations of parameter values for each step in the Procrustes superimposition. As there is arguably no objective criterion to determine which alignment best suits bivalve shells, we discuss the benefits and drawbacks of each approach and quantitatively compare the similarities of resulting alignments. Ultimately, we use this exercise to propose a best practice for aligning bivalve shells and comparing their shapes—a process that may be useful for workers in other, similarly disparate morphological systems that lack high degrees of point-based homology.

## METHODS

### Dataset

We adopt the style of previous approaches to studying bivalve orientation and use a dataset of morphological end-members to illustrate the effects of different alignment schemes (*e.g. Owen, 1952*; *Yonge, 1954*; *Stasek, 1963a*). Eleven species that represent most major body plans were selected to proxy the morphological and anatomical disparity across the evolutionary history of the Class (Table S1). Bivalves with highly reduced shells or those

(a) Commissure curve semilandmarks and their centroid

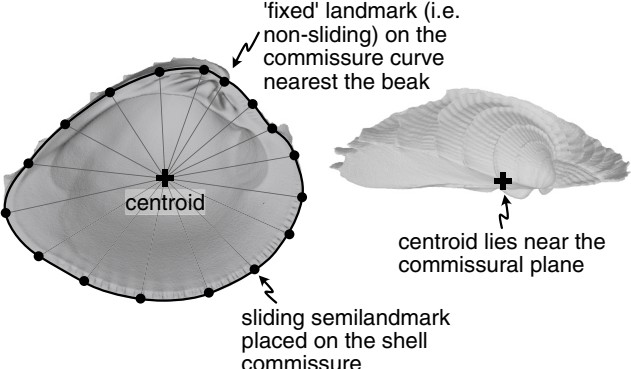

(b) Shell points and their centroid

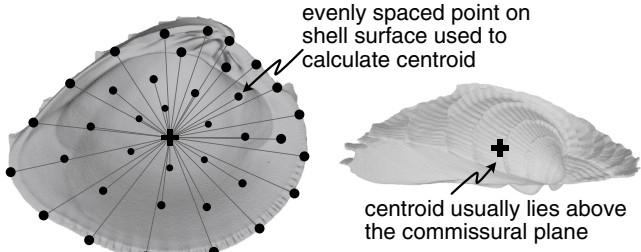

**Figure 2  Characterization of shell commissure and centroid.** (A) Representation of shell commissure curve, its centroid, and the semilandmarks used in the COMM orientation scheme. Analyses use 50 sliding semilandmarks on the commissure curve, but only a subset is shown here for clarity. (B) Equally spaced points on the shell surface placed using a Poisson Disc sampler (*Rvcg::vcgSample*, *Schlager, 2017*) and their centroid. The number and location of vertices on triangular meshes can vary, which strongly influences the calculation of centroid size. Analyses use 2,000 equally spaced points to minimize this issue (only a subset of those points shown here). Figured shell is *Chione elevata*.

that form part of a larger structure (tubes and crypts) are not directly analyzed here, but we consider their fit to the alignments in the Discussion.

One valve from an adult individual of each species was sampled from museum collections (Table S1). Nine of 11 individuals are equivalve, and because we do not examine details of dentition, their left and right valves are operationally mirror images of each other. The inequivalve taxa included here (*Pecten*, *Ostrea*) primarily differ in terms of valve width (height above the commissural plane); for the purposes of this analysis, the location of key features including the hinge area and adductor muscle scars are similar enough that using either valve gives a similar orientation. For larger-scale studies of morphology, we prefer to use the left valves of inequivalve taxa for homologous comparisons with equivalve taxa. Valves were scanned using micro-CT at the University of Chicago's Paleo-CT facility, and three-dimensional, isosurface, triangular-mesh models were created in VG Studio Max and cleaned in Rvcg (*Schlager, 2017*) and Meshmixer. Landmarks described below were placed using 'Pick Points' in Meshlab (*Visual Computing*

*Lab ISTI – CNR, 2019*). See Data Availability section for the meshes, landmarks, and R code necessary to reproduce the analyses described below.

## Aligning bivalve shells for comparative morphometrics
### Scaling

Procrustes superimposition scales objects to a common size, and three alternative scalings are considered here: (1) the **centroid size of the shell** (Fig. 2B), (2) the **centroid size of the commissure** (Fig. 2A), and (3) the **volume of the shell**. The centroid size of the shell reflects the 3D footprint of the shell, and the centroid size of the commissure reflects the size of the shell's growth front. Even though centroid size is mathematically independent of shape, the two are often correlated in biological data (*Zelditch, Swiderski & Sheets, 2012*:13); thus, shell volume—the amount of calcium carbonate—is also considered here because it may have a different association with shell shape than the other two size measures, possibly revealing other aspects of shape differences among the specimens (*e.g.* differential allometry among clades).

### Translation

After scaling, Procrustes superimposition translates objects to a common origin. Objects are typically 'centered' by subtracting the centroid of the landmark set (mean X, Y, and Z coordinate values per object) from each landmark coordinate, thus shifting the center of each landmark set to the origin (X = 0, Y = 0, Z = 0). Three points are considered for translation here: (1) the **beak** (Figs. 1A, 1D), (2) the **centroid of the shell** (Fig. 2B), and (3) the **centroid of the commissure** (Fig. 2A). Translation to the beak positions shells onto the homologous point of initial shell growth. Translation to the centroids of the shell or its commissure incorporate more information on the shape of the shell, with centering on the commissure adding an aspect of homology by positioning shells on their growth front. Operationally, Procrustes superimposition translates landmark sets to their respective centroids before minimizing their rotational distances, overriding any prescribed translations; the three translations above are therefore implemented after the rotation step (following the functionality in *Morpho::procSym Schlager, 2017*).

### Rotation

Rotation in Procrustes superimposition orients landmark coordinates to minimize their pairwise sum of squared distances. The five orientations discussed in the introduction were used for rotation. Because Procrustes superimposition uses Cartesian coordinates, two landmarks were placed on the mesh surface of a shell to mark the features used to define each axis as described in the subsections below (exact placement of landmarks on specimens in Fig. S1).

## Sagittal orientation
*Sagittal axis (SX).* This axis is the pole to the commissural plane (Figs. 1B, 1E). It is determined as the average cross product of successive vectors that originate at the centroid of the commissure and terminate at semilandmarks on the commissure curve

(visualization of fitting the commissural plane in Fig. S2). The distal direction is towards the exterior surface of the shell and the proximal direction is towards the interior surface.

## Anteroposterior orientation

*Hinge line (HL).* The hinge line is determined by landmarks placed at the two farthest apart articulating elements of the hinge area (Fig. 1). The landmarks are then designated as being anterior or posterior using the available discriminating features on the shell and can thus proxy the anteroposterior orientation. While not an 'axis' in the strict anatomical sense, we group the hinge line with the other anatomical axes below.

*Oro-anal axis (OAX).* The positions of the mouth and anus or proxies thereof are used to orient the oro-anal axis and thus the anteroposterior orientation. For dimyarian taxa, anterior and posterior ends of the axis are determined by landmarks placed on the dorsal-most edge of the anterior and posterior adductor muscle scars (Fig. 1A, the 'Type 2 adductor axis' of *Bailey, 2009*:493 after *Stanley, 1970*:19). For monomyarian taxa that have retained the posterior adductor muscle, the centroid of that adductor muscle scar is landmarked as the posterior end of the axis and shell features that reflect the position of the mouth are landmarked as the anterior end (*e.g.* the ventral notch of the anterior auricle in pectinids (Fig. 1D) or the beak in ostreids; *Yonge, 1954*:461). The axis is reversed in monomyarian taxa that have retained the anterior muscle (*e.g.* the protobranch Nucinellidae; *Glover & Taylor, 2013*:101).

## Dorsoventral orientation

*Maximum growth axis (GX).* The origin of shell growth at the beak is the dorsal landmark on the maximum growth axis and the point on the shell commissure with the greatest linear distance to the beak is ventral landmark (Figs. 1A, 1D).

*Orthogonal hinge line (oHL).* By definition, the orthogonal line to the HL (oHL) represents the dorsoventral axis, with the dorsal-most point nearest the beak.

*Orthogonal oro-anal axis (oAX).* By definition, the orthogonal axis to the OAX (oOAX) represents the dorsoventral axis, with the dorsal-most point nearest the beak.

## Commissure orientation

Landmarks were manually placed on the shell commissure with sufficient density to capture its shape (usually 25–50 landmarks were needed). These landmarks were then used to fit a three-dimensional spline on which 50 equally spaced semilandmarks were placed in an anterior direction (clockwise for left valves when viewed towards the interior surface, counterclockwise for right valves). The semilandmark on the commissure curve nearest the beak landmark was selected as the initial point for the curve (Fig. 2A). Semilandmarks were then slid to minimize bending energy and reduce artifactual differences in shape driven by their initial, equidistant placement (*Gunz, Mitteroecker & Bookstein, 2005*; *Gunz & Mitteroecker, 2013*; *Schlager, 2017*).

## Standardized axis points

The relative placement of axis landmarks varies greatly among specimens (Fig. S1). Such variable displacement in landmark positions can strongly influence the minimization of

**Standardization of axes defined by landmarks**

1. Landmarks placed to set axis orientation.

Hinge Line

Oro-anal Axis

Maximum Growth Axis

2. Initial points shifted to the centroid of the commissure.

3. Axis 'vectors' normalized to unit length, then duplicated and multiplied by -1 (dashed 'negative' vectors).

4. Axis 'startpoint' set as the terminal point of the negative vector and axis 'endpoint' set as terminal point of original vector.

**Figure 3** **Visualization of procedure used to standardize the orientation axes defined by landmarks.** Figured shell is *Chione elevata*.

the least-squares criterion during rotation, where the most variably placed landmarks tend to dominate the final alignment (see also the 'Pinocchio effect' in *Zelditch, Swiderski & Sheets (2012)*:67). Thus, axis landmarks were 'standardized' to equalize the contribution of each axis to the specimen's rotation (visualization of this process in Fig. 3). The vector defined by the two axis landmarks was shifted to the centroid of the shell commissure and normalized to unit length; the standardized axis points (we explicitly avoid calling them landmarks) were then designated by the terminal points of the unit vector and its negative. Standardized axis points result in alignments that better reflect the collective impacts of axis direction, rather than magnitude.

## Alignment and comparison of shape differences

Meshes and landmark sets for right valves were mirrored across their commissural plane and analyzed as operational left valves. This is a reasonable approach for equivalve taxa when analyzing general shell shape, *e.g.* of the interior or exterior surfaces, but homologous valves should be used for analyses that include inequivalve taxa as, by definition, their two shapes differ. Landmark sets were then scaled, centered and rotated, and translated (in that order) under all possible parameter combinations outlined in the preceding section, totaling 45 alignment schemes. Landmark coordinate values were scaled by dividing the landmark coordinates by a specimen's size (*i.e.* the centroid size of the shell points, commissure, or its volume). Scaled landmarks were then temporarily centered on the centroid of the commissure and then rotated *via* the respective orientation scheme using Generalized Procrustes superimposition (*Morpho:procSym*, *Schlager, 2017*); scaling during

this step was explicitly disallowed. Lastly, scaled and rotated landmarks were translated to one of the three target locations (*i.e.* the beak or centroids of the commissure semilandmarks or shell points).

Similarity of alignments was quantified using the metric distances between the shapes of interior shell surfaces, which were used to reduce the impact of exterior ornamentation on the differences in general shell shape. Sliding semilandmarks on the commissure and the interior surface of the shell were used to capture 'shape.' Initially, for the commissure, 50 equidistant semilandmarks were placed with the curve's starting point determined by the orientation scheme (*e.g.* starting at the semilandmark nearest the beak for the SX-COMM orientation, see details in Supplemental Text §2.3, Fig. S5); for the interior surface, semilandmarks were placed at proportionate distances along the dorsoventral and anteroposterior axes of each orientation scheme (5% distance used here, which results in 420 semilandmarks; see details and step-by-step visualization in Supplemental Text §2.3, Figs. S3–S5). Mixing the orientations of semilandmarks and rotation axes may be useful for comparing the interaction of growth and anatomical direction with shell shape, but this approach can result in unintuitive, and perhaps unintended, shape differences among specimens. After placement of equidistant semilandmarks, those on the commissure curve were slid to minimize their thin-plate spline bending energy and then used to bound the sliding of the surface semilandmarks (Fig. S5; *Gunz, Mitteroecker & Bookstein, 2005*; *Gunz & Mitteroecker, 2013*; implemented *via Morpho::slider3d*, *Schlager, 2017*). The final sliding semilandmark set consisted of 430 landmarks (50 points on the commissure plus the 380 points on the surface grid which do not lie on the commissure, *i.e.* the non-edge points; Fig. S5). Landmark coverage analyses may be used at this point to maximize downstream statistical power (*Watanabe, 2018*), but we relied on qualitative assessment of shape complexity and landmark coverage for the simple analyses conducted here.

For each of the 45 alignments, similarity in shell shape was calculated as the pairwise Euclidean distances of the sliding semilandmarks placed on the interior surface of the shell and the commissure. Identical shapes have a distance of zero. Pairwise distances between shapes for each alignment scheme were normalized by their respective standard deviations, making the distances between specimens comparable across alignments. The normalized pairwise distances in this 'alignment matrix' were then compared in three ways.

First, the alignment matrix was scaled and centered and Principal Components Analysis was conducted to visualize the individual and joint effects of treatments across alignments. Second, the effects of Procrustes superimposition steps on the scaled pairwise distances between specimens was modeled using a MANOVA with Type III sum-of-squares and residual randomization in a permutation procedure (as implemented by *RRPP::lm.rrpp* and *RRPP::anova.lm.rrpp*, *Collyer & Adams, 2018*, who describe this procedure as an ANOVA on either univariate or multivariate data). Each of the 45 rows in the analyzed matrix was a unique Procrustes superimposition treatment, or alignment—*i.e.* a combination of scaling, rotation, and translation—and each of the 55 columns was a scaled distance between a pair of the eleven specimens. Post-hoc tests of differences in the means of treatments within superimposition steps were conducted using *RRPP::pairwise* (*e.g.* the

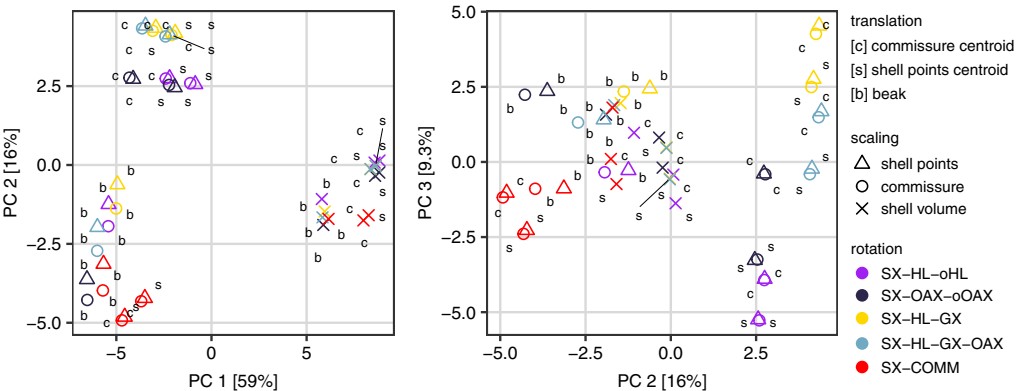

**Figure 4 Principal components analysis of the 45 Procrustes superimpositions (represented by points).** PCs 1–3 explain 84.3% of the total variation in the alignment model space.

differences in alignments when shells were scaled by volume or by the centroid size of the commissure). Differences in the predicted means of treatments within superimposition parameters were evaluated while holding the other superimposition parameters constant (*e.g.* the differences in predicted mean alignments of rotation schemes evaluated at the mean effects of translation and scaling, see *RRPP::predict.lm.rrpp*); these differences were visualized along the first two principal components of the predicted mean values with 95% confidence ellipses generated from the residual randomization procedure. These metric differences among alignments are informative for understanding the impacts of individual steps in the Procrustes superimposition, but visually comparing the orientations of shells is necessary to understand an alignment's fidelity to biological homology and/or analogy. Thus, visual inspection of alignments and 'hive diagrams' were used as a third means of comparing the scaled pairwise distances of specimens among selected alignments to a reference alignment, where scaling = centroid of commissure, rotation = HL-oHL, translation = centroid of shell commissure. All analyses were conducted in R v. 4.1.3 (*R Core Team, 2022*), details on all packages and their versions included in the supplemental code.

## RESULTS

All three Procrustes superimposition steps—translation, scaling, and rotation—impact the alignment of shells. Scaling by shell volume *vs.* the centroid size of the shell or the commissure primarily separates the alignment model space along PC 1, which explains 59% of the total variance (Fig. 4); scaling also has the largest standardized effect in the MANOVA of the alignment model space (*i.e.* the largest *Z* score in Table 1). PCs 1 and 3 explain 16% and 9.5% of the total variance in the alignment model space, respectively; together, these PCs show the clustering of alignments using the SX-COMM rotation, translation to the beak, as well as the similarity of alignments using translation to the centroids of the commissure semilandmarks and shell points, and some separation of the axis-based rotation schemes (Fig. 4). The apparently similar effects of rotation and translation in partitioning the alignment model space are also reflected by their similar

**Table 1 Results of the multivariate analysis of variance on the alignment model space (i.e. the scaled pairwise distances between specimens across the 45 Procrustes superimpositions).**

| Superimposition step | df | Sum of squares | Mean square | $R^2$ | F | Z | p |
|---|---|---|---|---|---|---|---|
| Translation | 2 | 30.6 | 15.3 | 0.11 | 13.7 | 4.4 | 0.001 |
| Scaling | 2 | 185.4 | 92.7 | 0.64 | 82.7 | 6.2 | 0.001 |
| Rotation | 4 | 34.7 | 8.7 | 0.12 | 7.7 | 4.8 | 0.001 |
| Residuals | 36 | 40.3 | 1.1 | 0.14 | | | |
| Total | 44 | 291.0 | | | | | |

effect sizes in the MANOVA (Table 1). As expected from the distributions of alignments within the PCA, scaling by shell volume produces a mean alignment of shells that is significantly different from scaling *via* the centroid size of the shell points or the commissure while the effects of translation and rotation are held constant (Fig. 5A, the later two do not produce significantly different alignments). Similarly, translation to the beak produces significantly different mean alignments from those translated to the centroids of the shell points or commissure (Fig. 5B). Most rotation treatments produce significantly different mean alignments from one another, but the alignments that incorporate combinations of the oro-anal axis and growth axis tend to produce similar alignments when translation and scaling are held constant (Fig. 5C).

# DISCUSSION

Choices of translation, scaling, and rotation can each impact the alignment of shells and thus their differences in shape. Below, we discuss the implications of those choices for analyses conditioned on aspects of the shell growth or function, and then briefly discuss the utility of certain alignment schemes for analyzing shell morphology across broad phylogenetic scales.

## Effects of translation

The choice of translation can change how differences in shell shape are interpreted. Translation to the beak, the lone homologous point across the class, allows the comparison of shell shapes conditioned on directions of growth from their origins (Fig. 6C). *Ensis* can be described as being posteriorly elongated compared to *Glycymeris*, or *Pecten* as 'taller' than *Pholas* from the beak to the ventral margin. Still, translation to the beak can exaggerate or bias the differences in 'pure' shell shape. For example, *Ensis* and *Tagelus* have greater distances between their shapes when translated to the beak than to the centroid of commissure semilandmarks (red line in Fig. 6D.iii). Their offset positions of the beak underlie this difference, which is interesting for analyses of growth *vs*. shape, but the shape of the shell, irrespective of its growth, is arguably the primary target of ecological selection (*Stanley, 1970*, *1975*, *1988*; *Vermeij, 2002*; *Seilacher & Gishlick, 2014*). Thus, measuring the morphological similarity of shells for studies of ecomorphology, trends in disparity, or evolutionary convergence would be best conducted using translation to their respective centroids of the commissure or shell surface (Figs. 6A, 6B); these two translations yield very similar alignments given the close proximity of their respective centroids (as shown by

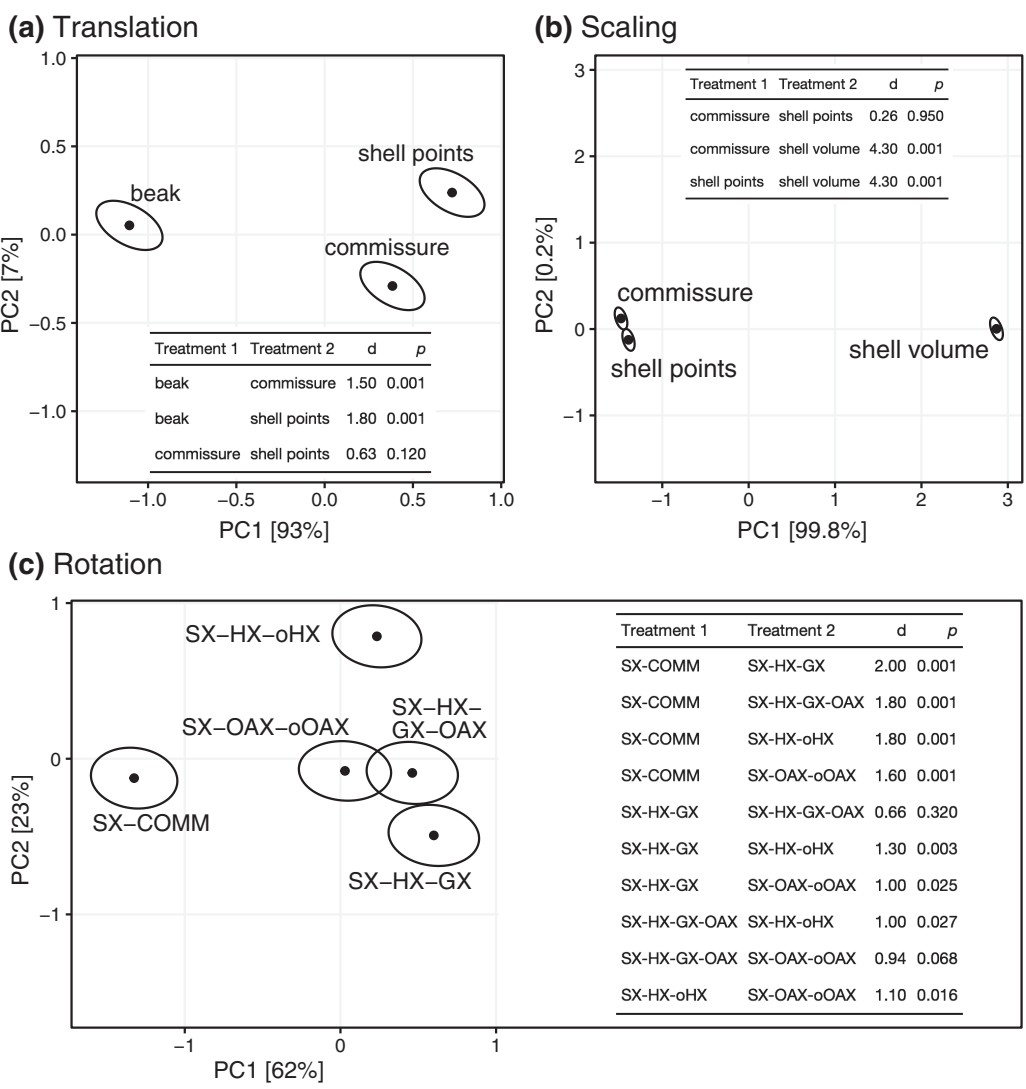

**Figure 5 Differences among treatments for each step of Procrustes superimposition.** Panels A–C show separate principal components analyses of predicted mean alignments for treatments within Procrustes superimposition steps while holding the effects of the other two. Points show mean predicted values and ellipses show 95% confidence boundaries. Each panel also shows a table of *post hoc* pairwise tests for differences in mean values among treatments from the MANOVA. The table header 'd' gives the distance between predicted mean values of treatments, and *p* values are derived from the *RRPP* sampling described in the Methods.

the pale colors linking specimens in Fig. 6D.ii; but note the small offset between the two centroids for the more irregularly shaped *Cuspidaria*). In general, it is best practice to translate shells to their respective centroids of the commissure semilandmarks or shell points when morphological analyses target differences in pure shell shape. Translation to the centroid of the commissure incorporates homology into the alignment *via* correspondence of the leading edge of shell growth.

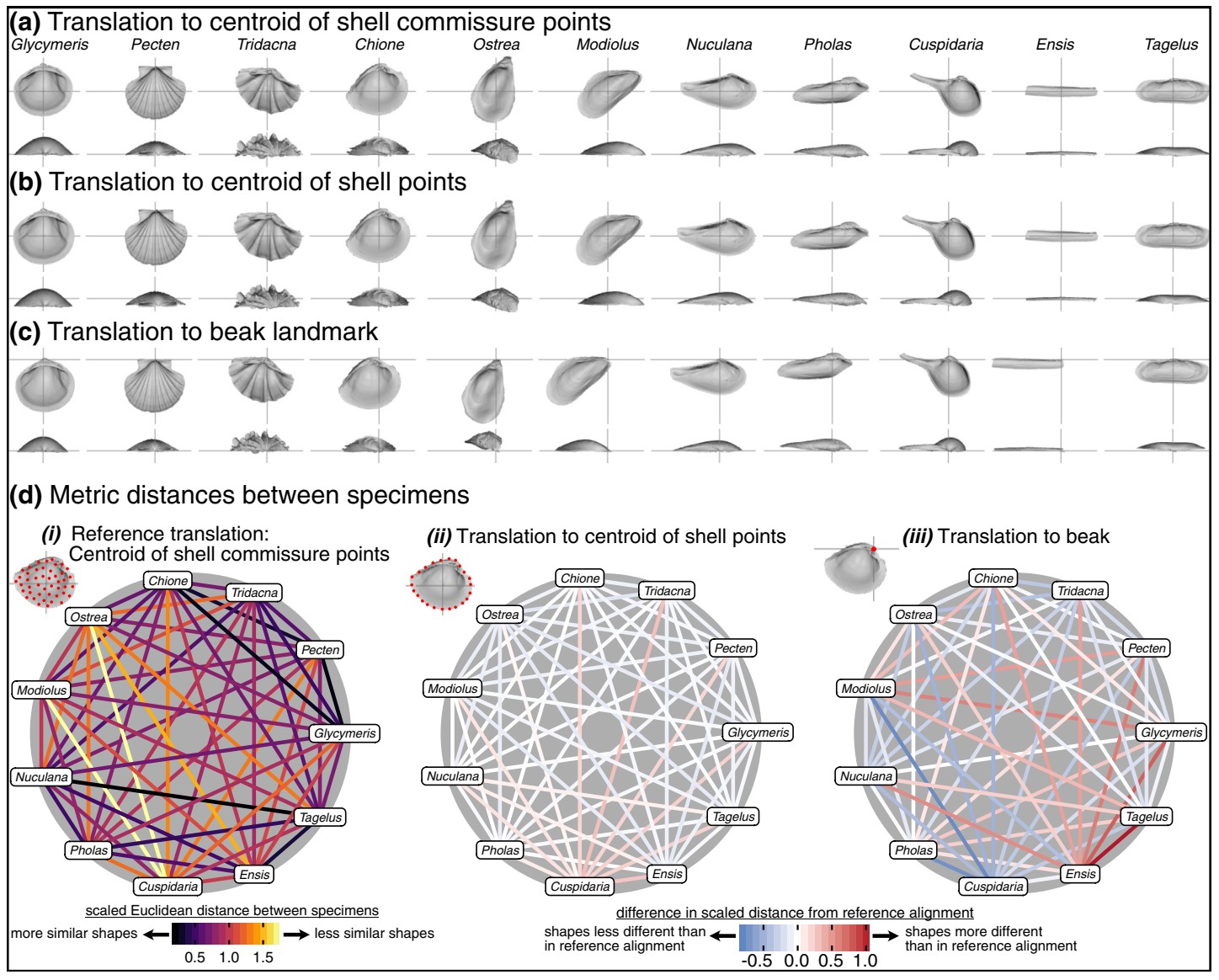

**Figure 6 Effects of translation on differences in shell shapes.** All shells are scaled to the centroid size of the shell points and rotated using the SX-HL-oHL scheme. For individual images of shells, the intersection of the gray line segments marks the origin of the Cartesian coordinate system and thus the operational 'center' of the shell. (A) Translation to the centroid of the 2,000 equidistant points placed on the mesh surface of the shell. (B) Translation of shells to the centroid of the 50 semilandmark curve along the shell commissure. (C) Translation of shells to the apex of the beak landmark, the initial point of shell growth. (D) (i) The scaled pairwise Euclidean distances of semilandmarks placed on the interior surface of the shell, scaled to the centroid size of the shell points and translated to the centroid of the shell commissure. 'Hotter' colors indicate greater relative distances between specimens. (ii–iii) The difference in scaled distance of specimens for the specified translation from the reference treatment in panel *i*. More saturated reds indicate an increase in scaled distance relative to the reference alignment; conversely, more saturated blues indicate a decrease in distance; white indicates no difference. For example, *Ensis* and *Tagelus* become more dissimilar in interior shell shape when translated to their respective beaks than when each are translated to their centroid of the commissure.

## Effects of scaling

Choice of scaling produces the largest absolute differences between alignments (Table 1, Fig. 4). Scaling by volume leaves particularly large residual differences in shape; the most voluminous shells are made extremely minute (*Pecten* and *Tridacna* in particular, Fig. 7C)

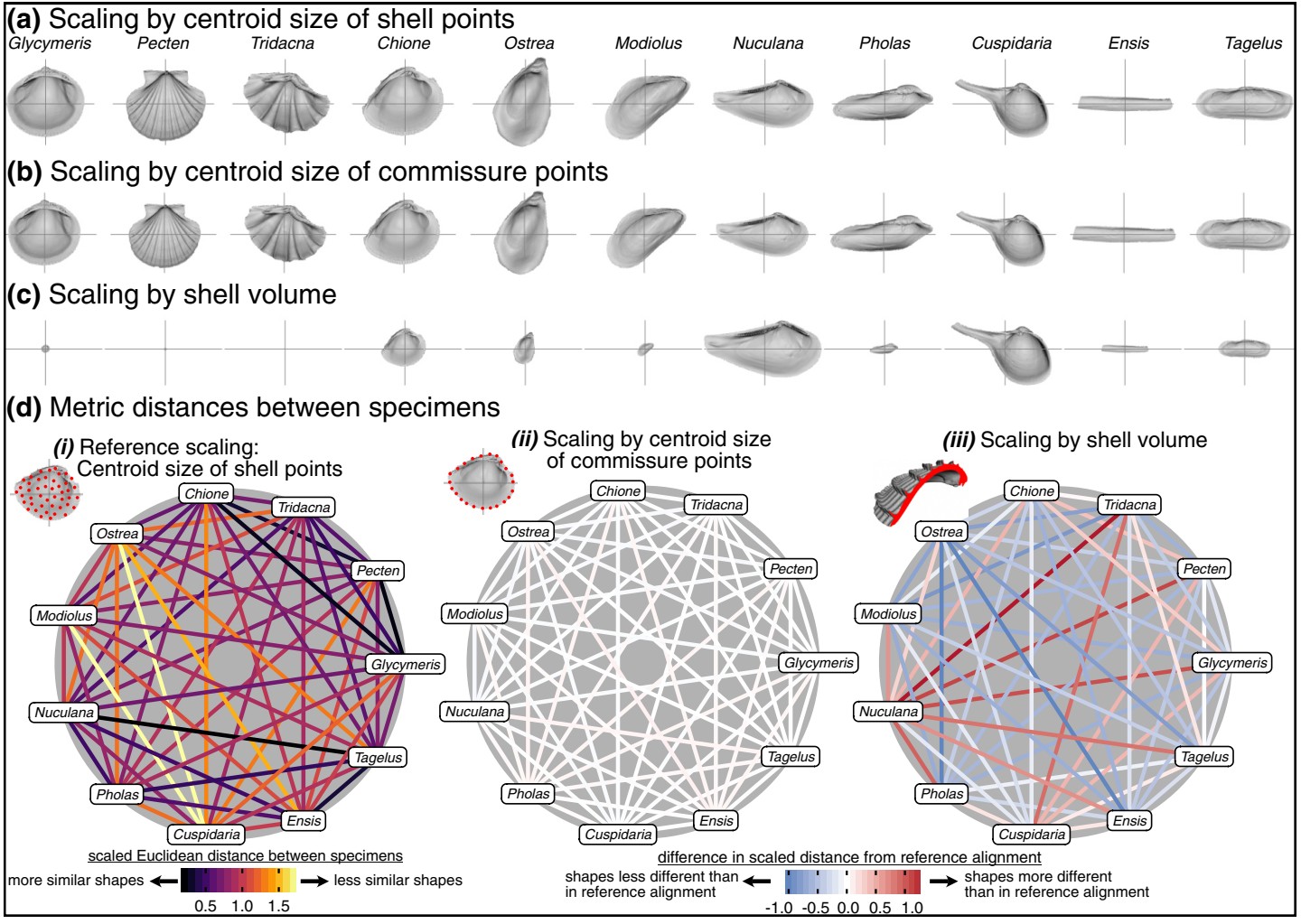

**Figure 7 Effects of scaling on differences in shell shapes.** All shells are translated centroid of the commissure semilandmarks and rotated using the SX-HL-oHL scheme. Compare differences in scaled sizes of specimens across rows, not columns. (A) Shells scaled by the centroid size of the 2,000 equidistant points placed on the surface of the shell mesh. (B) Shells scaled to the centroid size of the 50 semilandmark curve along the shell commissure. (C) Shells scaled by the volume of shell carbonate. (D) As in Fig. 6D but based on differences in scaling.

while the least voluminous shells become the largest (*Nuculana* and *Cuspidaria*). Scaling to logged shell volume does not alleviate these residual differences (results not shown), and, moreover, the aim of this scaling step is to remove the isometric relationship of size to shape, not its allometric one. The relative sizes of specimens are more similar when scaled to the centroid size of the commissure semilandmarks or the shell points (Figs. 7A, 7B). These two sizes are tightly correlated (Fig. S6) and thus produce very similar alignments (Fig. 7D.ii). For comparing differences in overall shell morphology in 3D, scaling by the centroid size of shell points best equalizes the isometric differences in size among specimens, thus concentrating the remaining differences in morphology to their shapes.

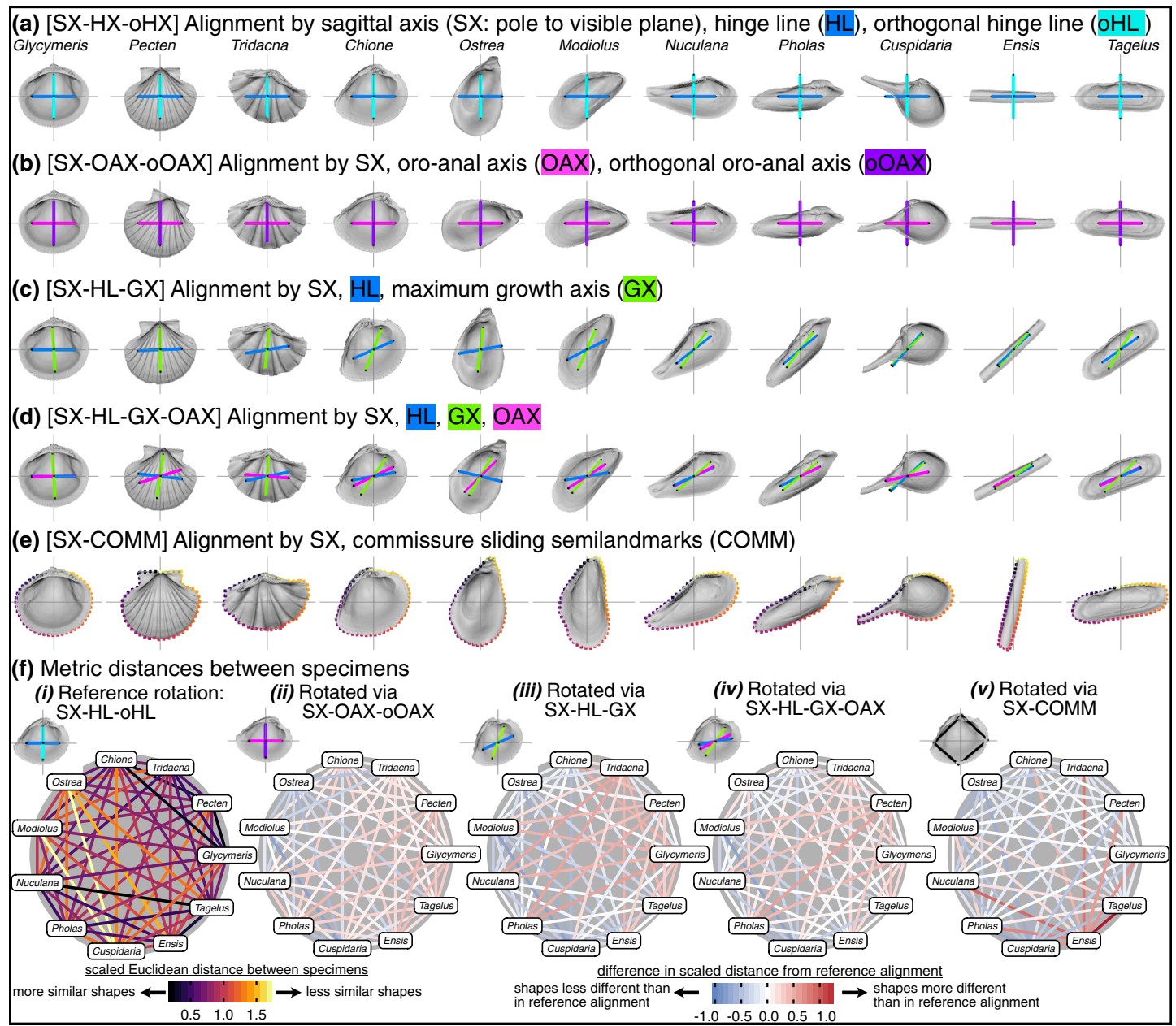

**Figure 8 Effects of rotation on differences in shell shapes.** All shells are translated centroid of the commissure semilandmarks and scaled to the centroid size of the shell points. Highlighted colors of panel titles correspond to axes plotted on shells. To facilitate relative comparisons of shell shape across columns, shells in each row were rotated such that the 'x' axis is parallel to the hinge line of *Glycymeris*; this is an ad-hoc, global rotation that does not change between-specimen differences in shell shape. (A) Shells rotated by their sagittal axis, hinge line, and orthogonal hinge line as the pseudo dorsoventral axis. (B) Shells rotated by their sagittal axis, oro-anal axis, and orthogonal oro-anal axis as the pseudo dorsoventral axis. (C) Shells rotated by their sagittal, hinge, and maximum growth axes. (D) Shells rotated by their sagittal, hinge, maximum growth, and oro-anal axes (E) Shells rotated by their sagittal axis and commissure semilandmarks. (F) As in Fig. 6D but based on differences in rotation. See Fig. S7 for a projection of shell shape differences along the first two principal components.

## Effects of rotation

Visually, rotation treatments can produce nearly orthogonal orientations of specimens (compare the orientation of the traditional shell length axis for *Ensis* and *Tagelus* between

SX-HL-oHL and SX-COMM, Figs. 8A, 8E; also reflected in the deep-red bar linking these two taxa in Fig. 8F.v and spacing of specimens on the first two PC axes in Fig. S7). Equilateral shells are aligned similarly across rotation treatments (compare orientations of *Glycymeris*, *Pecten*, and *Tridacna*, in Figs. 8A–8E and the less saturated lines connecting them in Fig. 8F.ii–8F.v). Differences in alignments become more pronounced among the more inequilateral shells (seen to a minor extent in *Chione* relative to *Glycymeris* and *Pecten*, but notably in *Modiolus, Pholas, Cuspidaria*, and *Ensis*). Thus, alignments of inequilateral shells tend to reflect a compromise between the often subparallel but not orthogonal orientations of their axes (most clearly seen in the changes to the orientation of *Modiolus*, *Pholas*, and *Ensis* relative to *Glycymeris* and *Pecten* in Figs. 8A–8F). Rotation by sliding semilandmarks on the commissure results in a similar alignment of most shells to the hinge line orientation (pale lines in Fig. 8F.v), but the relative shape differences of *Modiolus*, *Ensis*, and *Tagelus* indicate the importance and impact of the beak position. The commissure curve begins at the point nearest the beak, which affects the orientation of the surface semilandmark grid (see Fig. S5). Thus, in the SX-COMM treatment, the growth and therefore 'shape' of *Modiolus* and *Ensis* is more similar to the tall-shelled *Ostrea* than either are to the putative, similarly elongate *Pholas* and *Tagelus* (which themselves become more dissimilar in shape owing to the slight offset in their beak positions). Overall, rotation using the hinge axis and its orthogonal axis as the pseudo dorso-ventral axis is likely the best practice for most analyses of shell shape, as discussed below.

## Practical considerations for comparative morphology of bivalve shells

In biological systems with limited homology in a strictly point-based, geometric sense—and even in those with plenty of it—numerous approaches have been used to align specimens for morphological analysis. Homology-free approaches that rely on the geometric correspondence of points have been used to analyze the pure shape differences in structures such as mammal teeth and ankle bones (*Boyer et al., 2015*). However, the alignments of specimens using this approach can often mis-orient samples with distinctive morphology, and *post hoc* corrections can be needed to fully align the dataset (which has been our experience applying this method to disparate shells; see *Boyer et al., 2015*:258–261 for an overview of the procedure). Other "landmark-free" approaches including spherical harmonics can provide a more continuous characterization of shape across surfaces, but even this approach requires an initial alignment using homologous landmarks to produce biologically meaningful shape differences (*Shen, Farid & McPeek, 2009*:1009). Thus, testing ecological and evolutionary hypotheses of morphological differences among specimens still requires *a priori* determination of alignment.

The axis-based approach to alignment (Figs. 8A–8D) is useful both for its ability to encompass broad phylogenetic analyses of shell morphology and for its ability to combine extant and fossil data, the latter known almost exclusively from shells. All shell morphologies should fit within this scheme, including those with strong lateral asymmetry (*e.g.* rudists, see *Jablonski, 2020* and those with calcified tubes or crypts such as teredinids and clavagellids, *Morton, 1985*; *Savazzi, 1999*, each of which have identifiable valves with anatomical axes—whether to include the tubes and crypts as aspects of shell morphology is
a different debate). With increasing phylogenetic proximity, the number of point-based biological homologies is likely to increase, permitting more traditional approaches to specimen alignment (*Roopnarine, 1995*; *Márquez et al., 2010*; *Serb et al., 2011*; *Collins, Crampton & Hannah, 2013*; *Pérez, Alvarez & Santelli, 2017*; *Sherratt, Serb & Adams, 2017*; *Collins, Edie & Jablonski, 2020*; *Edie et al., 2022*; *Carmona, Lazo & Soto, 2021*). These shell-based axes and features (Figs. 8A–8E) are also useful for incorporating fossil taxa into analyses with extant taxa (*Yonge, 1954*; *Cox, Nuttall & Trueman, 1969*; *Stanley, 1970*; *Bailey, 2009*), but aspects of the internal anatomy remain crucial for orientation (*Stasek, 1963a*), especially the designation of the anterior and posterior ends. Fortunately, in many cases the anteroposterior axis can be determined from imprints of the soft body on the shell surface (*e.g.* the pallial sinus) or from other shell features (*e.g.* siphonal canals, pedal gapes). This necessarily variable and often idiosyncratic approach to defining the direction of anatomical axes may result in more digitization error than seen in traditional point-based geometric morphometrics. However, the impact of that error on analytical interpretations of shape similarity and variance will depend on the overall scale of shape disparity; for analyses of morphology across the class, the latter is likely to far exceed the former.

As for most analytical frameworks, comparisons of shell shape will require explicit definition of the alignment scheme and interpretation of any differences within those boundaries. Comparisons among different methods will be the most powerful approach to testing evolutionary hypotheses (see *Bromham, 2016* for the necessity of comparative analyses in historical science). For example, *Savazzi (1987*:298) follows *Seilacher (1970)* in suggesting that "the shape of an organism can be interpreted as the interaction of a number of factors, which can be grouped into the categories of constructional, functional and evolutionary constraints." Here, the empirical shape differences in constructional and evolutionary effects on shape could be analyzed *via* the commissure-orientation landmarking scheme (SX-COMM) and the functional effects *via* the hinge line landmarking scheme (SX-HL-oHL). Even more precise mapping of the association between shell shape and growth could be accomplished by placing semilandmarks using modeled growth parameters (after *Raup, 1966*, see application to brachiopod shells in *Polly & Motz, 2016*:89), or by digitizing ontogenetic growth segments (*Pérez & Santelli, 2018*). Comparison of the resulting empirical morphospaces would provide a means to quantify the many-to-one mapping of growth to form among bivalves (*e.g.* the similar, rectangular-shaped but differently constructed *Ensis* and *Tagelus* in Fig. 8), and would thus contribute to understanding of evolutionary pathways to convergence in shell shape (*Anderson, 2014*:31).

Each alignment presented here corresponds to an ecological, developmental, or evolutionary hypothesis for differences in shell form, but we do recognize a generally applicable approach that, to us, best reflects the decades of study of shell morphology: alignment *via* the sagittal axis, hinge line, and its orthogonal line as the pseudo-dorsoventral axis (SX-HL-oHL). Shell height, length, and width have been the principal measurements for analyzing differences in shape, and long-standing, taxon-specific 'rules' have become entrenched in the literature and therefore influence our

interpretations of the clade's evolutionary morphology (see discussion in *Cox, Nuttall & Trueman, 1969*:81–82 and the continued utility of these measurements in *Kosnik et al., 2006*). The SX-HL-oHL rotation tends to orient shells according to the defined axes of those linear measurements. Of course, precedent need not dictate the course of future work, but here, we find it reasonable to align this 'next generation' of shell-shape analyses with the long-standing conventions in the literature, if only for comparative purposes. This approach may not strictly adhere to the paradigm of Geometric Morphometrics (*Bookstein, 2018*), given that it relies on transformations of primary landmarks for the purpose of rotation, and because scaling and translation are not derived from those primary landmarks. Still, this axis-based alignment with biologically informed scaling and translation produces intuitive gradients in shell morphology that are grounded in homology. Further, using a landmark-based approach to characterize the shell surface permits the inclusion of other landmarks describing the shapes of additional shell features, such as the muscle scars and aspects of the hinge plate for analyses of partial disparity, modularity, and/or integration (*Edie et al., 2022*; see *Polly & Motz, 2016* and *Goswami et al., 2019* for examples in other biological systems).

## Applications to other model systems with accretionary growth

This approach of specimen alignment may be particularly relevant to other model systems in paleobiology and macroevolution that have accretionary-style growth: brachiopods and conchiferan molluscs such as gastropods, cephalopods, rostroconchs, etc.—each with limited point-based landmarks corresponding to biological homology. Of particular interest would be adapting the comparative morphological framework to test for differences among these groups in the disparity of their associations between form and growth. For example, gastropod shell shape modeled as a function of its growth parameters (*e.g. Collins et al., 2021*) could be compared to shell shape measured with respect to a functional axis or plane (*i.e.* the aperture). Further, the anteroposterior and dorsoventral directions of gastropods are apparent within the plane of the aperture, with the third dimension defined by the spiral trace of the aperture centroid. Unlike for bivalves, this alignment scheme would likely produce orientations that differ from the conventional spire 'up' and aperture 'forward' illustrations, but as such, may reveal new patterns of morphological variation. Given that Mollusca is the second most speciose phylum of Animalia and their disparity has so far hindered comparative analyses that test clade-wide evolutionary-morphology mechanisms, adopting more generalized alignment and shape characterization schemes can facilitate these phylogenetically broad approaches.

## CONCLUSIONS

The debate on how to align specimens is still relevant in the current era of morphometry, where comparisons of animal form are increasingly accessible in 2D, 3D, and even 4D (*Boyer et al., 2016*; *Olsen, Camp & Brainerd, 2017*; *Pearson et al., 2020*). No matter how shapes are compared, interpretations of their differences or variances should be with respect to an assumed anatomical alignment. For comparisons of disparate morphologies, particularly those that lack biological homology conducive to point-based landmarking,

alignments will likely require non-standard approaches so that shape differences do not depend on geometric correspondence alone. In bivalves, anatomical axes inferred from taxon-specific features offer a Class-wide approach to orientation. One set of axes in particular (SX-HL-oHL) coincides with historical approaches to their morphometry, while another offers new insight into the relationship between shell shape and shell growth (COMM-SX). Each solution is valid in its own way, as both relate to a specific evolutionary question.

## ACKNOWLEDGEMENTS

We thank the following museums and their staff for access to the specimens used in this study: Field Museum of Natural History, National Museum of Natural History, Natural History Museum, U.K., and M. K. McNutt and D. Wolfe for their support at an early phase in writing this article. We also thank D. B. Provete, D. E. Pérez, P. Milla Carmona, and F. L. Bookstein for thoughtful, constructive comments that significantly improved the clarity, precision, and purpose of this paper.

### Funding

This work was supported by the National Aeronautics and Space Administration (NNX16AJ34G), the National Science Foundation (EAR-0922156, EAR-2049627), and the University of Chicago Center for Data and Computing. The funders had no role in study design, data collection and analysis, decision to publish, or preparation of the manuscript.

### Grant Disclosures

The following grant information was disclosed by the authors:
National Aeronautics and Space Administration: NNX16AJ34G.
National Science Foundation: EAR-0922156, EAR-2049627.
University of Chicago Center for Data and Computing.

### Competing Interests

The authors declare that they have no competing interests.

### Author Contributions

- Stewart M. Edie conceived and designed the experiments, performed the experiments, analyzed the data, prepared figures and/or tables, authored or reviewed drafts of the article, and approved the final draft.
- Katie Susanna Collins conceived and designed the experiments, performed the experiments, analyzed the data, prepared figures and/or tables, authored or reviewed drafts of the article, and approved the final draft.
- David Jablonski performed the experiments, analyzed the data, authored or reviewed drafts of the article, and approved the final draft.

## Data Availability

The mesh models, landmark data, and code for reproducing analyses and figures are available at Zenodo: Edie, Stewart M. (2022). Specimen alignment with limited point-based homology: 3D morphometrics of disparate bivalve shells (Mollusca: Bivalvia) (2.1) (Data set). Zenodo. https://doi.org/10.5281/zenodo.6568044.

The mesh models are available at Morphosource (Project ID 000429826):
- https://doi.org/10.17602/M2/M429843
- https://doi.org/10.17602/M2/M429837
- https://doi.org/10.17602/M2/M429849
- https://doi.org/10.17602/M2/M429855
- https://doi.org/10.17602/M2/M429831
- https://doi.org/10.17602/M2/M429862
- https://doi.org/10.17602/M2/M429868
- https://doi.org/10.17602/M2/M429874
- https://doi.org/10.17602/M2/M429894
- https://doi.org/10.17602/M2/M429888
- https://doi.org/10.17602/M2/M429881.

## Supplemental Information

Supplemental information for this article can be found online at http://dx.doi.org/10.7717/peerj.13617#supplemental-information.

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
