# Peer review of "Specimen alignment with limited point-based homology: 3D morphometrics of disparate bivalve shells (Mollusca: Bivalvia)"

_PeerJ, doi:10.7717/peerj.13617_

## Round 0.1 · original submission · Major Revisions

I was able to secure three reviews on your manuscript. While two reviewers made only a few comments, R2 expressed a bit of concern with the overall GM approach used. However, I invite authors to take Bookstein's comments with a grain of salt and only incorporate his suggestions if agreed. That would change almost the whole paper, which I think is not warranted and not necessary.

It's rather unusual to see an Introduction with subheadings. I'm not even sure you need those. I'd highly suggest you to try to accommodate the content of the subheadings throughout the introduction and then close the introduction with the study goals. It's much easier to understand them this way. I have given some advice in the pdf attached.

Take a closer look at the statistical analysis. You definitely didn't use a PERMANOVA, but a three-way MANOVA coupled with RRPP. Improve that description

https://besjournals.onlinelibrary.wiley.com/doi/abs/10.1111/2041-210X.13029

Results and Discussion must be separated from each other to distinguish fact from interpretation.

·

Basic reporting

This paper addresses a problem well-known to morphometricians working with bivalves in a broad phylogenetic context, i.e., how to produce meaningful shape normalizations when a limited number of homologous points are available. The authors start by discussing anatomical features of the bivalve shell that have been traditionally used to define body axes and allow its standardized orientation. These axes and features are used to devise, implement and quantitatively compare a number of strategies for aligning the bivalve shell using 3D morphometrics (including a number of novel alternatives to standard procedures) and a sample of shells encompassing the diversity of body plans of the bivalvia. The paper closes with a discussion of the biological meaning of the differences found between approaches and the potential adequacy of the different strategies studied.

This is a very clear, well-written, self-contained study. The authors succeed in providing conceptual, historical and technical context for the reader, and the figures (and table) provided are fit for purpose. While the paper slightly departs from the standard sections of the journal, I think its structure is appropriate for addressing the issue at hand. The authors share both the raw data and the code they used, and include supplementary information that is very useful to delve into the specifics of their methodology.

Experimental design

This piece of research falls well within the scope of PeerJ. The issue being tackled (which is relevant primarily for morphometricians and bivalve researchers, but more generally for any researcher working with structures with few or no discrete homologue points) and its importance are adequately explained and framed. Throughout the article, the authors explore strategies specifically devised to mitigate the problem addressed, making an effort to insert these within the corresponding body of knowledge. Their anatomical discussion (and related literature) and methodological descriptions (and associated R code) stand out as particularly valuable contributions to the topic (in fact, I was wondering if the approach to morphometric quantification used here has been published elsewhere–if not, maybe the related supplementary text and figures could be relocated into the main text).

The rationale of the paper is sound and the explanations of the morphometric procedures undertaken are clear and very detailed, needing only minor corrections, clarifications and/or additions (see attached PDF). Regarding reproducibility, I was not able to run the script “Edie_etal_bivalignment” beyond line 403 because the function “points_along_plane” was missing (also, the script with helper functions does not run beyond its line 472 when called with source, I had to do it manually). However, as far as I could see this material is excellent and would be extremely useful for anyone interested in understanding and replicating this approach. It would be great if the authors could fix these problems and provide a fully functional script.

Validity of the findings

I was able to download and check both the raw data and code (stored in Zenodo), although as mentioned earlier I was not able to replicate all the results reported in the paper due to the lack a function. In any case, all the results and findings of the article seem valid and stemming from sound reasoning and adequate application of methodology (my only concern in this regard would be the results from perMANOVA, as I am not sure the function chosen by the authors is the right alternative to perform this particular analysis; see comments in page 16 of the attached PDF). The conclusions arrived at are all supported by the results and discussion, and are adequately aligned with the research goal.

Pablo Milla Carmona

·

Basic reporting

English is perfect. Literature references are insufficient as noted in my review. The paper is not self-contained (it fails to review the axioms of GMM).

Experimental design

The paper is original. Its question, however (how to adapt geometric morphometrics to bivalve macrotaxonomy) is not meaningful.

Validity of the findings

The authors' basic conclusion, as per the Abstract sentence my review quotes, is correct, but the paper should have had the courage to say that any other expectation represents a profound misunderstanding of the actual logic of geometric morphometrics, which has no role to play here.

Additional comments

See attached .pdf of a full review text.

·

Basic reporting

The article has been written in clear, unambiguous and very professional English. Also, the article includes very nice figures and a table, and a very useful amount of supplementary data.

The authors share links to the raw data.

Figures are relevant and properly labeled and described.

The article is well structured, including acceptable and standard sections. However, I suggest a more clear separation between results and discussion (see comment in ms). This could improve how you show the extension of your results for a broad audience and future applications. So, this does not mean separation in two sections (Results-Discussion), I propose rephrasing some sentences about discussion (those referring the focus of your study, recommendations for use, goals, extension of the problem, etc.) of the sub-section “Practical alignments for bivalve shells”. Also, the issue about extension of the methodology to other model systems is not developed in the discussion and I hope to see it in the discussion (see comments in ms).

Experimental design

This is an original, well defined, relevant research, focusing in a becoming problem associated to the methodologies used today.

This research is clearly defined in its focus, and the contribution to the field is clearly defined and delimited.

Methodology are clearly described with sufficient details, is rigorous and replicable from the information and data provided. So, fine details of the methodology are beyond my knowledge and expertise.

Validity of the findings

Results of this contribution are novel, and to assess with pertinent tools a problem related to methodologies with a broad use.

As I said above, the extension of the results are briefly mentioned.

Decisions are not subjective, they are based on results. All needed data to replicate are provided and are available.

Conclusions are well stated.

Additional comments

The article assesses a very interesting problem concerning to a broad used methodology today, in palaeontology and morphology. Details of the methodoloy are beyond of my expertise, but the methodological bases and procedures are well worked. One of the goals of this work is to link biological/anatomical basis of different approaches with methodological usages, based on the revision of morphological nomenclature and the morphology itself. I hope to see this article published.

The following lines are related to my comments on the ms:

The introduction are short and concise, but some words about the disparate body plans (examples) are needed. Also, a more detailed explanations about the advantages and disavantages on the use of shells would be adequate. They considered living and extant taxa as a continuous and I agree with this. So, these results are important to palaeontologists and morphologists equally. The use of shells is an advantage for it, and for the facility to access, availability in collections, etc.

Figure 1 needs indications of specific parts. Only "a" and "b" are insufficient. This figure contains a lot of information explained in the text, and more indications help to the reader.

Dataset needs abbreviatures of collections used. This is referred to acknowledgements on Methods section but is not totally provided, more collections are indicated in the Table S1. I suggests they be included in the supplementary text.

Results and Discussion section is correct, but the sub-section "Practical alignments for bivalve shells" has some details: some lines seems to be recommendations or preferences. I encourage to authors to be more clear about this. Please provide recommendations and/or arguments on the results.

I suggest to include in the Results/Discussion section some lines about the extension of our phylosophy of alingment to other model systems. This is mentioned at the abstract and the conclusions, but not developed in the core of the article.

A line in the supplementary text refers to "Dijkstra's algorithm" but this is not referenced.

---

## Round 0.2 · Minor Revisions

The same two reviewers from the previous round commented again on the paper. They mentioned minor improvements in the suppl mat, R script, and a sentence that needs further clarification to avoid confusion between correlation and allometric relationship. Also, correct the part when you mention you conducted an ANOVA, which is actually a MANOVA (your response variable is multivariate; the ANOVA function in R is generic and does any testing depending on the input data).

Finally, PeerJ uses a structured abstract. I highly recommend authors to adhere to this template.

I'd kindly ask the authors to take a final look at those comments and provide a revised version of the text.

·

Basic reporting

The original version of this manuscript was an already clear, self-contained, nicely written study tackling a relevant issue in bivalve morphometrics (although of potential interest to researchers of other groups too), providing adequate figures, table and literature, and potentially useful digital materials and code (although the latter needed fixing). The current version of the draft improve most of these aspects further, as the necessary clarifications and corrections have been incorporated by the authors.

Experimental design

All the aspects in need of revision have been taken care of. I have only one fairly minor comment on the manuscript regarding lines 285-289:

After reading your responses and corrections, I realized that by 'correlation' you were referring to allometric variation (which es what Zelditch 2012:13 is taking about too) and not 'methodological independence' as I thought initally. I think that for the sake of clarity you should keep it simple and just be explicit about allometric variation (for example you could just modify the original sentence to mention allometric variation instead of correlation, e.g., “Shell volume–the amount of calcium carbonate–may be associated with allometric patterns different from those associated with the other two size measures, and therefore reveal other aspects of shape variation” or something along those lines.)

Aside from that, the R code is now almost fully functional (in line 62 of the script the closing parenthesis of “if” is misplaced and in line 1979 there is a comma lacking, although these are of course minor details that can be fixed easily by those checking this material by themselves), which is great for people interested in replicating this approach.

Validity of the findings

The only issue I was worried about (the PERMANOVA/Procrustes ANOVA one) has been clarified.

·

Basic reporting

As I mentioned in my previous review, this work is clear an unambiguous, well written and with a correct language. This reviewed version answered my questions and comments and expand some previously confussed points.

Separation between Results and Discussion, and expansion of the discussion about the extension of the methodologies are satisfying developed. Added sentences and references contribute to explain the methodologies and usages, and help to understand the topic for a wider audience (and add relevance to this research).

The new version is more concise and more complete and the figures now acompannied better to the main text.

Experimental design

I have not new comments about experimental design.

Sentences and explanation given to this topic are clearer than the previous version. I am happy with this new version.

Validity of the findings

As I mentioned in my previous review, this research shows novel results, referred to a known problem for whose use morphometrical tools with frequency (a broad distributed field). So, the conclusions are very well established.

Additional comments

Modifications suggested were well answered and new addition contributes to a more clear manuscript. I agree with the responses to all my comments or suggestions.

Because the subpanel labels on Fig. 1 I was able to more easily follow the flow of the manuscript. Thanks for this addition.

Expantion on abbreviatures of museum collection are correct, but the Table S1 have more abbreviatures than the mentioned in the Acknowledgement section. My previous missing comment (sorry for this error) in Table S1 indicates this point. May be the Table S1 could provide these data. Collection material is the primary basis of this study in my opinion, and the proposed and discussed methodologies are very useful for future collection-based research.

New section "Applications to other model systems with accretionary growth" adressed with my previous comment about the extension of the problem and dicussion. I was happy to read it.

Newly detailed rephrased references in the text to Comparative Morphometrics instead Geometric Morphometrics approach seems to me a very good assess to this work, expanded and illustraiting in a better view your work.

---

## Round 0.3 · accepted · Accept

Thank you for making these final amendments into the manuscript. I consider it adequate to be published as is now.
The function RRPP::lm.rrpp builds linear models both for uni- and multivariate response data, while running the generic function 'anova' on an object created with RRPP::lm.rrpp returns an anova *table* with F statistics and such. It works the same with models built with both uni- and multivariate response data.
Thank you for considering PeerJ as a venue for your interesting work.